# Unbend, correction of local beam-induced sample motion in cryo-EM images using a 3D spline model

**Lingli Kong[1]\*, Ximena Zottig[1,2], Johannes Elferich[1,2], Nikolaus Grigorieff[1,2]\***

[1]RNA Therapeutics Institute, University of Massachusetts Chan Medical School, Worcester, United States; [2]Howard Hughes Medical Institute, University of Massachusetts Chan Medical School, Worcester, United States

## eLife Assessment

This paper describes Unbend - a new method for measuring and correcting motions in cryo-EM images, with a particular emphasis on more challenging in situ samples such as lamellae and whole cells. The method, which fits a B-spline model using cross-correlation-based local patch alignment of micrograph frames, represents an **important** tool for the cryo-EM community. The authors elegantly use 2D template matching to provide **convincing** evidence that Unbend outperforms the previously reported method of Unblur by the same authors. Comparison to alternative programs for motion correction shows smaller gains, but with interesting differences between data sets.

**\*For correspondence:**
lingli.kong1@umassmed.edu (LK);
niko@grigorieff.org (NG)

**Abstract** The exposure of frozen biological samples to the high-energy electron beam in a cryo-electron microscope commonly leads to beam-induced sample motion and distortions. Previously, we described *Unblur*, software to correct for beam-induced motion based on the alignment of full frames in a movie collected during the beam exposure (Grant and Grigorieff, 2015). Here, we present *Unbend*, extending *Unblur* by accommodating more localized sample bending and distortions using a 3D cubic B-spline model. *Unbend* is integrated into our *cis*TEM software with a new local motion visualization panel. We processed movie frames from various in situ sample types, including whole cells, lamellae, and cell lysates, to analyze motion behavior across different specimen types. To quantify the improvement in high-resolution signal, we utilized the 2D template matching method to search large ribosomal subunits from the motion-corrected micrographs. Overall, the signal-to-noise ratio of detected particles improved by 3–8% across different samples compared with full-frame aligned micrographs, while the number of detected target particles increased by up to ~300%. Furthermore, we processed micrograph montages to study motion patterns across an entire sample, revealing considerable variance in distortion scale within the same sample, suggesting a complex underlying mechanism.

## Introduction

Cryo-electron microscopy (cryo-EM) enables structural biologists to observe diverse biomolecules in their native environment. The molecules are preserved in amorphous ice, and in the most favorable cases, their 3D structure can be reconstructed at near-atomic resolution. An important factor determining the attainable resolution is high-quality micrographs, containing high-resolution signal of the molecules. This high-resolution signal is attenuated by radiation damage and sample motion during the exposure (beam-induced motion [BIM]). To minimize this attenuation, images are commonly recorded as a series of movie frames, each containing a fraction of the total electron exposure (*Brilot*

*et al., 2012*; *Campbell et al., 2012*; *Li et al., 2013*). The frames are then filtered according to the radiation damage suffered by the sample and averaged into a single micrograph. We have previously developed the program *Unblur*, which is part of *cis*TEM (*Grant et al., 2018*), to perform alignment of full movie frames and exposure filtering (*Grant and Grigorieff, 2015*). However, as discussed in the following, full-frame alignment cannot correct for local distortions and bending of the sample under the electron beam.

While full-frame alignment is often sufficient to achieve 3D reconstructions at near-atomic resolution, more complicated patterns of sample deformation have been observed that cannot be corrected in this manner. A simple model describes the deformation as a drum-like motion (*Brilot et al., 2012*; *Zheng et al., 2017*; *Naydenova et al., 2020*), which leads to differential sample motion across the field of view. The primary driver of this motion and deformation is thought to be strain that develops in the sample during rapid freezing (*Naydenova et al., 2020*; *D'Imprima and Kühlbrandt, 2021*).

Different algorithms have been developed to correct micrographs and tomograms for more complicated motion patterns (*Li et al., 2013*; *Scheres, 2014*; *Abrishami et al., 2015*; *Grant and Grigorieff, 2015*; *Rubinstein and Brubaker, 2015*; *Zheng et al., 2017*; *Zivanov et al., 2019*; *Tegunov and Cramer, 2019*; *Střelák et al., 2020*). The widely used software *MotionCor2* employs polynomial equations to fit drum-like motion (*Zheng et al., 2017*). *Warp* (*Tegunov and Cramer, 2019*) and *Cryo-SPARC* (*Punjani et al., 2017*) use a spline-based model, which provides more flexibility in representing different types of distortion. More complicated distortion models may increase the resolution of 3D reconstructions of particles extracted from the corrected micrographs or tomograms. However, 3D reconstruction requires multiple image processing steps and often involves additional manual decisions, making it difficult to directly measure the impact of local distortion correction. Thus, it has not been quantified how great the improvements by local distortion correction are, nor how different types of samples, often exhibiting different degrees of distortion, benefit from it. To fill this gap, we set out to develop a tool to model and visualize local motion and to quantify the improvements in the micrographs upon distortion correction.

In this paper, we introduce new movie alignment software, *Unbend*, along with a comprehensive description of a new distortion correction model based on 3D cubic splines that is designed to apply pixel-wise correction of local motion. *Unbend* features two alignment steps: a full-frame alignment similar to *Unblur* (*Grant and Grigorieff, 2015*), which is part of *cis*TEM (*Grant et al., 2018*), and patch-based alignment to generate reference points to initialize the 3D spline model. The spline model is then refined by optimizing a summed patch-wise cross-correlation loss function. The shift amount for each pixel is computed using the refined model and applied to the micrograph to generate a corrected micrograph with improved high-resolution signal. We also wrote *shift_field_generation*, a program that reads the refined spline parameters generated by *Unbend* to plot pixel-wise shift fields for each movie frame.

We demonstrate the model's effectiveness using a variety of cellular sample types, including whole cells (both bacterial and mammalian), lamellae (yeast and mammalian cells), and cell lysates, highlighting how local motion varies across specimens. By tracking the displacement of each patch from the first to the last frame, we measured the amount of motion in these different samples. Using the pixel-wise shift fields of the last frame with respect to the first frame, we quantified the amount of deformation and calculated the total equivalent strain (overall magnitude of volumetric and deviatoric deformation, *Malvern, 1969*; *Belytschko et al., 2000*) and the Von Mises equivalent strain (magnitude of deviatoric deformation, *Hill, 1950*). By integrating these metrics with 2D template matching (2DTM) (*Rickgauer et al., 2017*), we directly quantified the effects of local motion in the micrograph and assessed the improvement in the 2DTM signal-to-noise ratio (SNR) following distortion correction. Our results show that *Unbend* improves the 2DTM SNR by 3–8% and increases the number of detected targets by up to 300% in a dataset, depending on the type of sample being imaged.

## Distortion modeling, correction, and verification

The alignment and local motion-correction pipeline proposed in this work consists of four stages. (1) We estimate full-frame (global) shifts for each movie frame. (2) We partition the globally aligned frames into patches and align each patch stack across frames to obtain per-patch shift trajectories. (3) We fit these patch-wise shifts with the 3D spline deformation model introduced here. (4) We refine the spline parameters by optimizing a cross-correlation-based objective, and use the resulting deformation field

to warp and sum frames to produce the final motion-corrected micrograph (*Figure 2—source data 1*). Detailed descriptions of each stage and the spline model are provided below.

## Full-frame alignment

During the acquisition of cryo-EM data, various types of sample motion can occur, including BIM and mechanical movement of the sample due to stage instabilities. Correcting for this motion is an important step in obtaining high-resolution 3D reconstructions of the imaged molecules. In this work, we address more complicated patterns of sample motion by separately performing full-frame alignment and local distortion correction due to sample deformation.

For full-frame alignment, we closely follow our previous version of *Unblur* (*Grant and Grigorieff, 2015*) using alignment in three iterations with increasing resolution. The shift for each frame is initially determined using the cross-correlation map between the current frame and the sum of all other frames. Given the low electron exposure per frame, single-frame alignment can be susceptible to noise and misalignment. A large B-factor, which can be adjusted by the user, is applied during this step to suppress high-resolution noise. Sample drift due to mechanical instabilities of the cryo stage is expected to be slowly varying in speed and mostly along a fixed direction. This allows the full-frame shifts to be smoothed using a Savitzky-Golay (SG) filter, which effectively models sample drift while preserving molecular features in the image, and without introducing potential image distortions. The full-frame alignment and trajectory smoothing are sufficient in many cases to achieve 3D reconstruction at near-atomic resolution (*Grant and Grigorieff, 2015*; *Oldham et al., 2016*; *Bartesaghi et al., 2018*).

In some cases, the shifts determined for individual frames deviate significantly from the smoothed trajectory. This could be due to noise but may also reflect properties of the sample or stage, and hence these outliers could represent real physical motion of the sample. Therefore, solely relying on the smoothed curves means that the shifts derived from cross-correlation are not directly applied in the correction, even if they represent real physical motion. To allow potential discontinuities in the motion trajectories in our new approach, this smoothing function is optional and is deactivated by default in *Unbend*. Instead of using the SG-filtered shifts, we compare the raw shifts to the smoothed curve and identify those shifts that deviate by more than 1.5 times the interquartile range (IQR) from the curve. These outlier shifts are further aligned using a B-factor twice that of the initial alignment (the default B-factor for the initial alignment is 1500 Å$^2$). If shifts remain significantly different after this adjustment, we accept them as the true shifts; otherwise, we apply the new shift values. This strategy helps us detect and correct potential misalignments caused by weak signal in individual frames while preserving the information contained in the raw shifts about real sample motion that does not fit a smooth trajectory.

## Patch alignment

To correct for local sample distortion, patch-based methods are widely utilized. By aligning small patches across the entire micrograph, these methods can accurately capture local motion. Increasing the number of patches and reducing the patch size can not only provide more accurate local information but also limit the amount of signal per patch, leading to noisier alignments. Users can set the patch size and patch numbers based on their preference. In our default settings, to balance these factors, we set a patch size of 1024×1024 pixels when the output pixel size is <0.5 Å, and 512×512 pixels when the output pixel size is between 0.5 and 2.0 Å. For output pixel sizes >2.0 Å, the 512×512 pixel patch size is scaled by a factor equal to the output pixel size and rounded up to the nearest multiple of 16 to maintain computational efficiency.

The default number of patches along the *x* and *y* dimensions, denoted as $NP_x$ and $NP_y$, are determined by rounding up the values of image dimensions divided by patch size. This configuration minimizes overlap between neighboring patches, enabling more accurate capture of local motion. Users may specify fewer patches along each dimension, resulting in larger patches that capture more signal to ensure complete coverage of the micrograph without gaps. Conversely, increasing the number of patches does not reduce their size; instead, it increases the percentage of overlap between adjacent patches, preserving the default patch size to ensure that each patch retains sufficient signal for robust alignment.

Assuming that full-frame alignment has effectively removed the average motion in each frame, the residual motion detected by patch alignment is primarily attributed to beam-induced sample distortion. We therefore approximate the shifts determined for a given patch across movie frames (a patch stack) as a smooth trajectory. For each patch stack, the alignment is first estimated by calculating the cross-correlation between an individual frame and the average of all other frames, after which the shift trajectory is smoothed using an SG filter.

To identify patches with unreasonable shift sets $\{(x_{pi,fi}, y_{pi,fi})|f_i \in \{1, 2, \ldots, NF\}\}$, where $NF$ is the total number of frames, $pi$ denotes the patch stack, and $x_{pi,fi}$ and $y_{pi,fi}$ represent the shift amounts along the $x$ and $y$ axes for frame $fi$ in patch stack $pi$. We calculate the standard deviation of the shift differences between neighboring frames:

$$\sigma_{fi}(pi) = std\left(\Delta S_{fi}(pi)\right),\tag{1}$$

where

$$\Delta S_{fi}(pi) = \sqrt{\left(x_{pi,fi+1} - x_{pi,fi}\right)^2 + \left(y_{pi,fi+1} - y_{pi,fi}\right)^2}.\tag{2}$$

Outliers in the set $\{\sigma_{fi}(pi) \mid pi \in \{1, 2, \ldots, NP\}\}$, with $NP = NP_x \times NP_y$ as the total number of patches, are detected using the 1.5×IQR upper bound criterion. Outlier shifts are then replaced with the shift sets from their nearest-neighbor patches. This procedure effectively detects potential misalignments caused by low signal in individual patches and provides a robust initialization for the subsequent modeling of sample distortion.

## Sample distortion modeling and correction

Patch-wise shifts, $\{(x_{pi,fi}, y_{pi,fi}) \mid pi \in \{1, 2, \ldots, NP\}, fi \in \{1, 2, \ldots, NF\}\}$, represent motion at the patch centers. To generate a distortion-corrected micrograph, these center shifts must be interpolated to obtain the shifts of every pixel in each movie frame. This requires an interpolation model, for which we employ spline functions to describe the pixel-wise shift distribution across the frames.

### Spline model construction

In this study, the spline model is constructed using uniform bicubic B-splines to describe shifts in the image plane ($x$ and $y$ axes), and uniform cubic B-splines to constrain shifts across frames ($z$ axis/electron exposure accumulating direction). B-spline curves and surfaces comprise sequences of low-degree polynomial segments, which help avoid high-degree global polynomials and preserve numerical stability. For both cubic and bicubic B-splines, the model guarantees $C^2$ continuity, i.e., it is twice continuously differentiable. This property allows for smooth transitions between polynomial segments while preserving local control, since changes in a control point only affect its neighboring region. As a result, the model provides sufficient flexibility to capture a wide range of motion trajectories.

Each segment of the spline curve $S_c(t)$ is a linear combination of basic B-spline functions $B_{i,d}(t)$, weighted by neighboring control points $Q_i$ (**Farin, 2001**):

$$S_c(t) = \sum_{i}^{i+d} Q_i B_{i,d}(t).\tag{3}$$

For degree $d$, the blending functions $B_i^d(t)$ are defined recursively by the Cox-de Boor algorithm:

$$B_i^0(t) = \begin{cases} 1, & t \in [t_i, t_{i+1}) \\ 0, & t \notin [t_i, t_{i+1}) \end{cases},\tag{4}$$

$$B_i^d(t) = \frac{t - t_{i-1}}{t_{i+d-1} - t_{i-1}} B_i^{d-1}(t) + \frac{t_{i+d} - t}{t_{i+d} - t_i} B_{i+1}^{d-1}(t),\tag{5}$$

where $t \in [0, 1]$ is a normalized parameter along the spline curve. For cubic B-splines, $d = 3$. A cubic B-spline segment can also be written in matrix formulation for better computational efficiency:

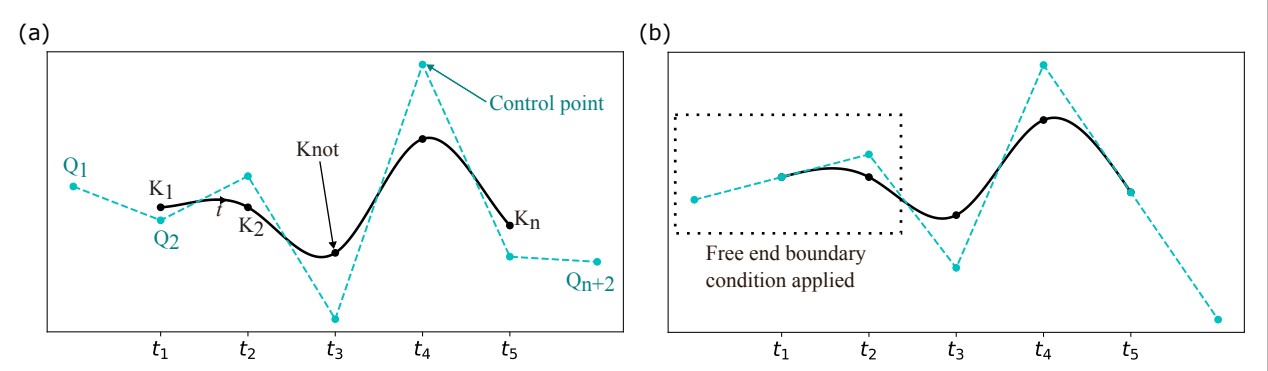

**Figure 1.** Uniform cubic spline curve. (**a**) Diagram showing the core components of a cubic B-spline curve, such as knots, control points, and the uniform knot positions $t_i$. (**b**) Diagram showing the free-end boundary condition for a cubic B-spline.

$$S_c(t) = \frac{1}{6} \begin{bmatrix} t^3 & t^2 & t^1 & 1 \end{bmatrix} \begin{bmatrix} -1 & 3 & -3 & 1 \\ 3 & -6 & 3 & 0 \\ -3 & 0 & 3 & 0 \\ 1 & 4 & 1 & 0 \end{bmatrix} \begin{bmatrix} Q_i \\ Q_{i+1} \\ Q_{i+2} \\ Q_{i+3} \end{bmatrix}. \tag{6}$$

Likewise, a bicubic B-spline surface $S_s(u_i, v_j)$ follows an analogous matrix formulation:

$$S_s(u, v) = \frac{1}{36} \begin{bmatrix} v^3 & v^2 & v^1 & 1 \end{bmatrix} M Q_{grid} \begin{bmatrix} u^3 \\ u^2 \\ u^1 \\ 1 \end{bmatrix}, \tag{7}$$

where $M$ is the same 4×4 blending-function matrix, and $Q_{grid}$ is the 4×4 arrangement of control points $\{Q_{i+k,j+l}\}$. The indices $i,j$ specify the segment positions along the $y$ and $x$ axes, while $u, v \in [0, 1]$ are the normalized parameters along those axes.

## Control points and knots

Each cubic B-spline curve segment is defined by four control points, with each control point influencing up to four adjacent segments, whereas each bicubic B-spline surface requires 16 control points. As shown in *Figure 1a*, control points are not located directly on the curve. The endpoints of curve segments and the four corners of the surface segments are referred to as 'knots'. Adjusting the control points alters the curve's shape, while the knots of a uniform spline are evenly distributed. To determine a B-spline curve/surface, we only need to know the knot parameters. The relationship between control points and knots for a curve or surface segment can be expressed as (*Agrapart and Batailly, 2020*):

$$K = \frac{1}{6}\Phi Q \,(\text{curve}), \quad K = \frac{1}{36}\Phi \,(\text{surface}), \tag{8}$$

where $\Phi$ is the passing matrix determined by the Cox-de Boor algorithm and the chosen boundary conditions. In this work, we adopt free-end boundary conditions, allowing the spline to extend linearly beyond the boundary points (*Figure 1b*). This prevents artificial curvature at the edges, maintaining the natural flow of the surface without imposing strict derivative constraints. For a cubic spline, the free-end boundary condition is implemented as follows:

$$Q_{k-1} - 2Q_k + Q_{k+1} = 0, \quad k \in \{2, n+1\}, \tag{9}$$

and for bicubic splines, the same condition applies along the edges. At the corners, the boundary condition is:

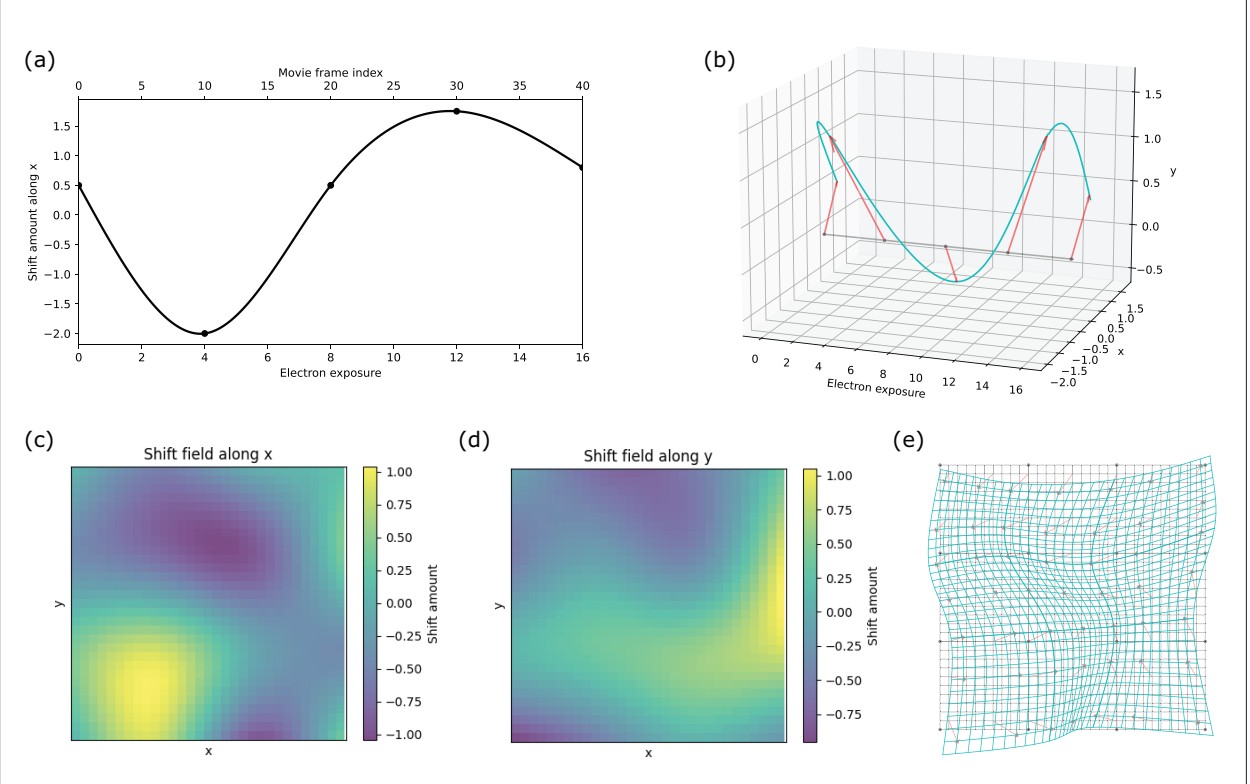

**Figure 2.** 3D spline model knot grid construction and shift field generation. (**a**) Diagram showing the position of knots along the exposure accumulation direction and the shift amount along $x$ for each movie frame. (**b**) A 3D diagram showing the cubic B-spline along the exposure accumulation direction, with red arrows representing the shifts where the knots are located (black dots on the gray line), and the cyan curve showing the spline curve for interpolating the shifts for each frame. (**c, d**) Shift field along $x$ and $y$ showing the shift amount of each pixel obtained from the bicubic B-splines on 30×30 pixels frame (for demonstration only). (**e**) Schematic image grid before (gray grids) and after (cyan grid) applying the shift fields in (**c, d**). The red arrows represent the shift of patches. Black dots represent knot positions.

The online version of this article includes the following source data for figure 2:

**Source data 1.** Unbend movie alignment and local motion correction algorithm.

$$Q_{k-1,l-1} - 2Q_{k,l} + Q_{k+1,l+1} = 0, \quad k \in \{2, n+1\}, l \in \{2, m+1\}, \tag{10}$$

where $m$ and $n$ are the number of knots along $x$ and $y$ axes, respectively.

## Knot grid configuration

Our model assigns $NK_x$, $NK_y$, and $NK_z$ as the number of knots along the $x$, $y$, and $z$ axes, respectively. By default, $NK_x$ and $NK_y$ are set to two-thirds of the patch number in each dimension, with a minimum of four knots, and evenly distributed along $x$ and $y$. Along the $z$ axis, knots are spaced at every 4 e⁻/Å² of exposure, forming a 3D grid together with the knots along $x$ and $y$. Since shifts occur in both $x$ and $y$, we maintain two sets of knot parameters, $K_x$ and $K_y$. Given these knots, we can interpolate a dense field of shifts for every pixel in each movie frame, thereby correcting local distortion in each frame.

*Figure 2* illustrates the knot grid and the construction of the 3D spline model. In *Figure 2a*, the spline curve for shifts along the $x$ direction is shown along the exposure accumulation direction. In this case, for a movie with 40 frames and a total exposure of 16 e⁻/Å², a knot is assigned every 10 frames. Once the shift amounts at the knots are determined, the cubic spline model, as described above, is used to compute the shift along $x$ for each movie frame. By combining these shifts with the shifts along the $y$-direction, we can obtain the full spline curve thread as shown in *Figure 2b*. For a model with $NK_x$, $NK_y$ knots along $x$ and $y$, we will generate $NK_x \times NK_y$ of such 3D splines. Thus, for each movie frame, we will have a 2D grid of knots for generating the bicubic spline surface, represented by the black points in *Figure 2e*. *Figure 2c and d* shows the shift fields along $x$ and $y$, respectively, for a

single movie frame derived using the bicubic splines. *Figure 2e* displays the image before (gray grid) and after (cyan grid) warping, based on the shift fields shown in *Figure 2c and d*. For better visualization, we amplified the distortion in these figures.

## Parameter refinement

The initial refinement of knot values relies on a least-squares error minimization approach. Given the parameters $K_x$ and $K_y$ for the $x$- and $y$-axis, respectively, the spline model can initialize the per-patch shifts for each frame $\left(x_{spline}\left(pi, fi; K_x\right), y_{spline}\left(pi, fi; K_y\right)\right)$. We estimate $K_x$ and $K_y$ by minimizing the following least-squares loss function:

$$L_1\left(K_x, K_y\right) = \sum_{pi=1}^{NP} \sum_{fi=1}^{NF} \left( \left[x_{pi,fi} - x_{spline}\left(pi, fi; K_x\right)\right]^2 + \left[y_{pi,fi} - y_{spline}\left(pi, fi; K_y\right)\right]^2 \right). \tag{11}$$

Minimizing $L_1$ aligns the spline model with the observed per-patch shifts and yields the patch stack alignment result $\left\{\left(x_{pi,fi}, y_{pi,fi}\right) \mid pi \in \{1, 2, \ldots, NP\}, fi \in \{1, 2, \ldots, NF\}\right\}$. Because this initial fit relies solely on local per-patch information, a second round of refinement is done to incorporate global information. To this end, we define the following loss function:

$$L_2\left(K_x, K_y\right) = -\sum_{pi=1}^{NP} \sum_{fi=1}^{NF} CC\left(I_{fi}^{shift}\left(pi; K_x, K_y\right), \overline{I}_{-fi}^{shifted}\left(pi; K_x, K_y\right)\right), \tag{12}$$

where $CC$ denotes the cross-correlation function between a single frame in a patch stack $I_{fi}^{shifted}\left(pi; K_x, K_y\right)$ and the 'leave-one-out' average of the remaining frames in the same patch stack:

$$I_{-fi}^{-shift}\left(pi; K_x, K_y\right) = \frac{1}{NF - 1} \sum_{\substack{fi' = 1 \\ fi' \neq fi}}^{NF} I_{fi'}^{shifted}\left(pi; K_x, K_y\right). \tag{13}$$

Since patches with stronger signal yield larger cross-correlation values, they contribute more heavily to $L_2$, thereby reducing the impact of noisier patches. The optimization algorithm L-BFGS, provided by the *dlib* package (*King, 2009*), is used to update parameters $K_x$ and $K_y$, producing a refined spline model that better captures the observed local movement. With these optimized parameters, pixel-wise shifts are computed, and bilinear interpolation is applied to calculate the densities at each output pixel, thereby generating the final motion-corrected movie frames. Averaging these frames produces the final corrected micrograph. As shown in *Figure 2e*, after applying the model-based shifts, areas near the edges may extend beyond or fall short of the original micrograph's pixel grid. In such cases, we pad the under-extended area with the average density of the edge and trim the over-extended area to fit within the boundaries of the micrograph.

## 2DTM provides a one-step verification for motion correction

2DTM was originally developed to identify biomolecules in images of crowded cellular environments with high precision in both location and orientation. The true-positive detection is based on the 2DTM SNR at each target site (*Rickgauer et al., 2017*; *Sigworth, 2004*). More specifically, at each image location $p$, we compute cross-correlation values $CC\left(p, t\right)$ between the micrograph and the template over all sampled orientations $t$. We then convert these values to z-scores per location $z\left(p, t\right)$, using the mean and standard deviation of $\{CC(p, t)_t\}$ across orientations:

$$z(p, t) = \frac{CC(p, t) - \mu_p}{\sigma_p}, \tag{14}$$

where $\mu_p$ and $\sigma_p$ are the mean and standard deviation of the cross-correlation values across orientations at location $p$, respectively. The 2DTM SNR is then defined as the maximum z-score:

$$2DTM\,SNR\left(p\right) = max_t z\left(p, t\right). \tag{15}$$

A detection is called when $SNR(p)$ exceeds a threshold $z_{th}$, which is determined based on the complementary error function $r_f = \frac{1}{2} erfc\left(\frac{z_{th}}{\sqrt{2}}\right)$, where the expected false positive rate $r_f$ is calculated by assuming there is one false positive detection (1-FP criterion).

Because the 2DTM SNR is computed by correlating template projections with the micrograph, it provides an unbiased measure of how effective the molecular signal is preserved in the distortion-corrected micrograph. In the present study, we will use both the number of detectable particles and their 2DTM SNR as metrics for evaluating the performance of our distortion correction method.

We applied our model to a range of in situ samples commonly studied in structural biology, including whole cells (bacteria and the edges of mammalian cells), lamellae (yeast and mammalian cells), and cellular lysate. Ribosomes, which can be detected with high accuracy by 2DTM using *cis*TEM (**Grant et al., 2018**), were chosen as the primary targets in the following experiments. Furthermore, we also tested the performance of our new algorithm on DeCo-LACE micrograph montages (**Elferich et al., 2022**).

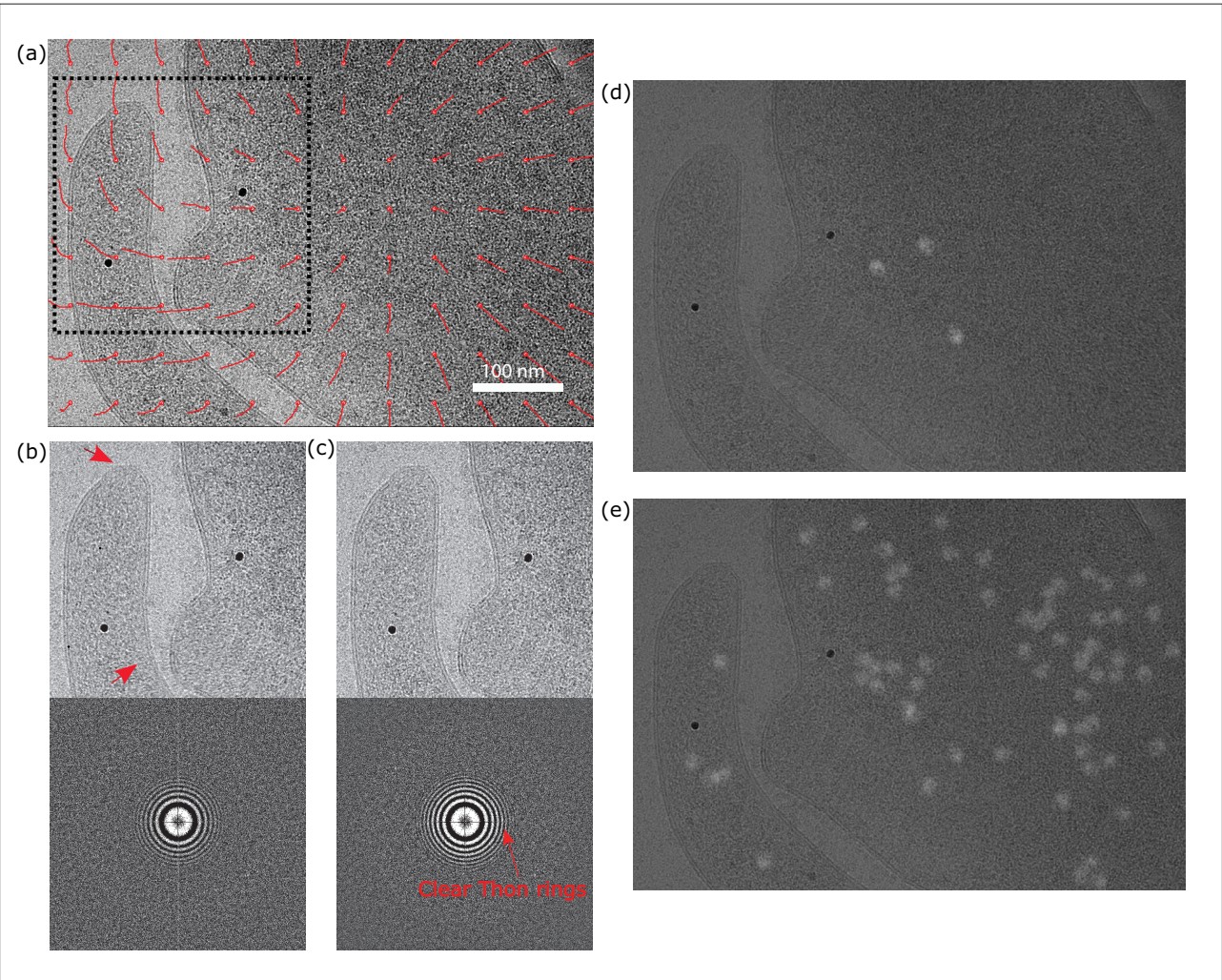

**Figure 3.** A micrograph of whole-cell *M. pneumoniae*. (**a**) Motion trajectory panel in the *cis*TEM graphical user interface (GUI). (**b**) The cropped area shown in (**a**) without local distortion correction, experiencing blurring due to the local distortion (upper) and the power spectrum (lower). (**c**) The cropped area (upper) and the power spectrum of the local distortion-corrected micrograph (lower). (**d**) Detected particles (white) using 2D template matching (2DTM) on the full-frame aligned micrograph. (**e**) Detected particles (white) using 2DTM on the local-distortion corrected micrograph.

## Results

### High-resolution information recovery

Using the new graphical user interface (GUI) for motion correction in *cis*TEM, users can readily visualize local motion by enabling the trajectory option. *Figure 3* illustrates an example micrograph acquired from *Mycoplasma pneumoniae* whole-cell samples. In this case, a vortex-shaped motion is observed in the region where two cells are closely adjacent to each other. This type of spatially varying, non-rigid deformation would not be well captured by simple, low-order global models, while the use of a continuous 3D spline representation in this work can interpolate smoothly across both space and time. The patch trajectories (shown in red) are magnified by a factor of 30 for clarity. *Figure 3b and c* provides zoomed-in views of the boxed area and the corresponding power spectra generated from the micrograph after local distortion-corrected alignment versus full-frame alignment, respectively. For movies exhibiting the highest estimated motion in each sample type in this work, the recovery of Thon rings at higher spatial frequencies is more apparent (see *Figure 4—figure supplements 1–5*). *Figure 3d and e* shows the detected particles (white projections) using 2DTM in the micrograph processed with and without local distortion correction, respectively. The larger number of detected 2DTM targets demonstrates that the blurring caused by the local distortion has been significantly reduced by our *Unbend* program. This can also be seen in the power spectra calculated from the micrograph, exhibiting more pronounced Thon rings at higher resolution after local distortion correction (*Figure 3b and c*).

### Local motion across different samples

We benchmarked our motion-correction model using three types of in situ specimens: whole cells, lamellae, and cell lysates. For whole cells, we used a frozen-hydrated *M. pneumoniae* dataset (*O'Reilly et al., 2020*) previously employed for the development of 2DTM with *cis*TEM (*Lucas et al., 2021*). To enable comparison across different sample types and geometries, mammalian cells were also included; specifically, immortalized *Cercopithecus aethiops* (African green monkey) BS-C-1 kidney

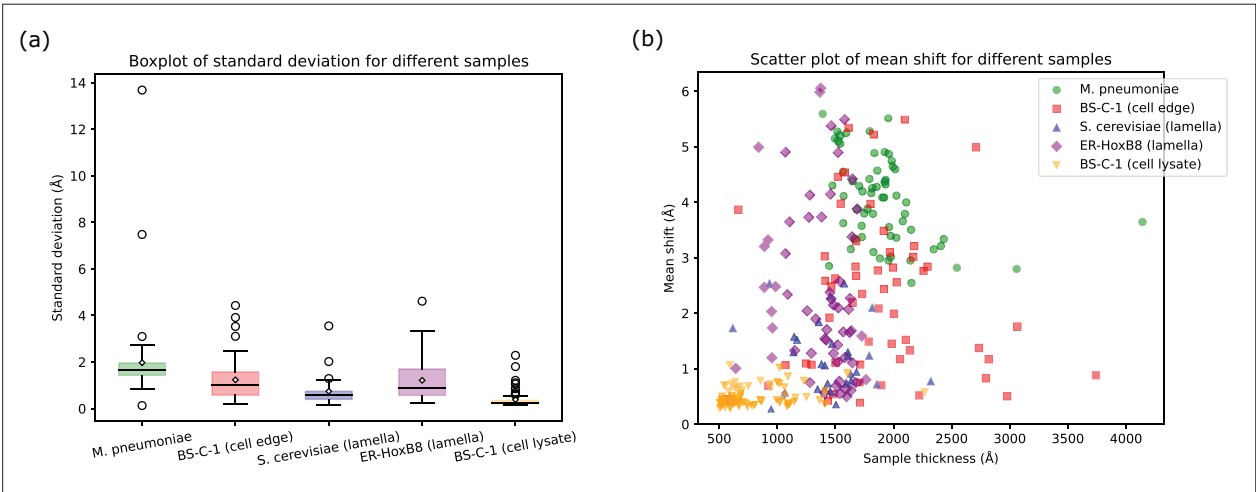

**Figure 4.** Patch shift in different types of samples. (**a**) Box plots of the standard deviation of patch motions calculated for micrographs of different types of samples. (**b**) The mean patch shifts for each micrograph, colored by sample type.

The online version of this article includes the following source data and figure supplement(s) for figure 4:

**Source data 1.** Summary statistics of per-micrograph mean patch shifts by sample type.

**Source data 2.** Motion and equivalent strain summary table for micrographs shown in *Figure 4—figure supplements 1–5*.

**Figure supplement 1.** *M. pneumoniae* sample.

**Figure supplement 2.** BS-C-1 cell edge sample.

**Figure supplement 3.** *M. musculus* ER-HoxB8 cell lamella sample.

**Figure supplement 4.** *S. cerevisiae* lamella sample.

**Figure supplement 5.** BS-C-1 cell lysate sample.

epithelial cells (see Materials and methods). Lamella datasets originated from two eukaryotic organisms – *Saccharomyces cerevisiae* and immortalized *Mus musculus* ER-HoxB8 cells – both previously analyzed by 2DTM (*Lucas et al., 2022*; *Elferich et al., 2022*). The cell lysate sample was derived from BS-C-1 cells, as described in Materials and methods.

To quantify the local movement, we measure the displacement of each patch's last frame relative to its first frame in a micrograph $\left\{ sh_{pi} \mid pi \in \left\{ 1, 2, \ldots, NP \right\} \right\}$, where

$$sh_{pi} = \sqrt{\left( x_{pi,NF} - x_{pi,1} \right)^2 + \left( y_{pi,NF} - y_{pi,1} \right)^2}. \tag{16}$$

Since the full-frame alignment is assumed to remove the global average motion, the patch motions/shifts ($sh_{pi}$) primarily represent the BIM that reflect the sample's deformation, measuring how much the patch centers are displaced from their initial positions. To characterize the extent of local motion, we compute the maximum, mean, and standard deviation of these shifts for each micrograph. We find that cell lysate exhibits the least local motion, whereas the *M. pneumoniae* samples demonstrate substantially larger local motions, with maximum displacements of over 55 Å. *M. musculus* cell lamella, and BS-C-1 cell edge samples also show considerable local motion (up to 20 Å). As indicated by the standard deviation data in *Figure 4*, samples with large average patch motion also exhibit greater variability in local motion across a micrograph. Such pronounced local motion will significantly attenuate high-resolution information in the micrograph, making its correction essential for high-resolution cryo-EM experiments.

To assess whether local motion correlates with specimen thickness, we estimated the sample thickness using CTFFIND5 (*Elferich et al., 2024*) and plotted the mean shift against sample thickness for each micrograph (*Figure 4b*). For consistency, all micrographs were binned to 1.5 Å/pixel during alignment, and outliers shown in *Figure 4a* are excluded for better pattern visualization. BS-C-1 cell edge samples span a broad thickness range (50–400 nm), and their mean displacements likewise cover the full range of observed shifts. *M. pneumoniae* are generally thicker (150–400 nm), and their mean displacements also distribute in the upper range (2.5–6 Å). In contrast, cell-lysate specimens are thinner (50–150 nm) and exhibit the least local motion, and even at the upper end (100–150 nm) of this range exhibit mean shifts below 1 Å. However, lamellae of similar thickness show mean displacements from <1 to >6 Å. These results indicate no direct dependence of local motion on sample thickness; rather, motion magnitude is highly related to the sample type and preparation method.

## Pixel-wise deformation quantification using equivalent strain

To quantify pixel-wise shifts within each movie frame, we developed *shift_field_generation*, a program that reconstructs the per-pixel displacement field of any given frame by interpolating the spline parameters exported by *Unbend* during movie alignment. Using the first frame as a reference, we compute the displacement of the final frame and derive two complementary strain measures: Von Mises equivalent strain, which measures distortional (shape-changing) effects, and total equivalent strain, which captures both volumetric (area-changing) and distortional components (see Materials and methods). Both measures range from 0 to infinite, with 0 designating no deformation, and

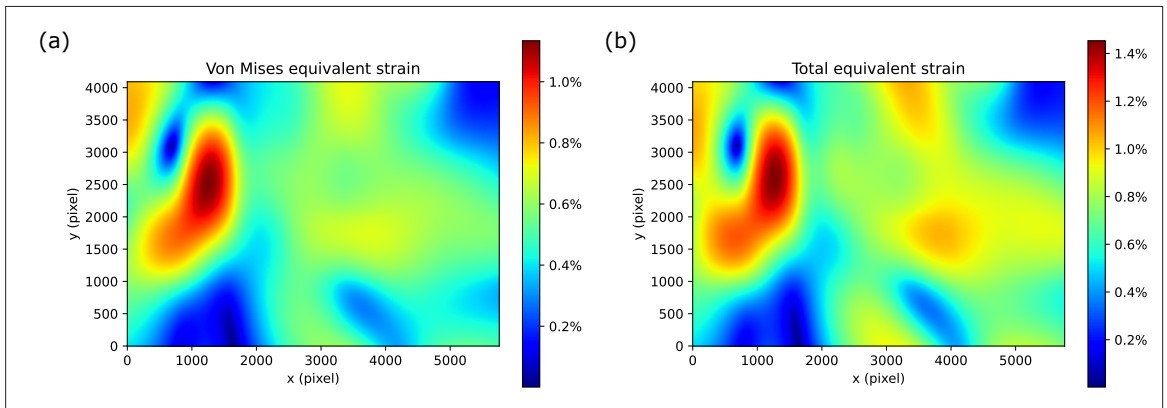

**Figure 5.** Magnitude of deformation correction of the micrograph shown in *Figure 3a*. (**a**) Von Mises equivalent strain and (**b**) total equivalent strain.

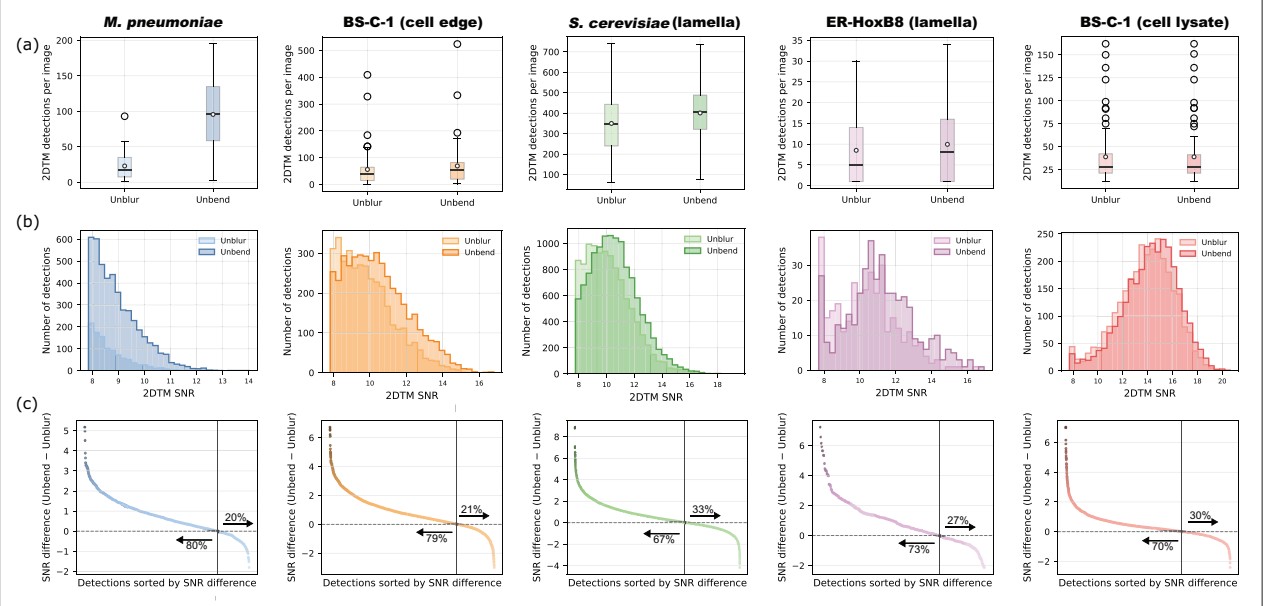

**Figure 6.** Evaluation of local motion correction using *Unbend* across different sample types. (**a**) Box plots displaying the distribution of the number of 2D template matching (2DTM) detections per micrograph. (**b**) Histograms of the 2DTM signal-to-noise ratio (SNR) of detected 2DTM targets. *Unbend* improves the overall distribution of SNR values for all sample types. (**c**) 2DTM SNR difference between *Unbend* and *Unblur*. Across all sample types, *Unbend* led to improvements in either the number of detected targets or 2DTM SNR values, or both.

provide scalar quantities that condense a multidimensional strain (or stress) state to equivalent single-dimensional values.

*Figure 5* presents strain maps (per pixel) for the example micrograph shown in *Figure 3*. The Von Mises (Green-Lagrange strain) map (*Figure 5a*) highlights a localized shear-dominated 'vortex' region where strain exceeds 1%, indicating significant non-isotropic deformation, which is consistent with the rotation motion shown by the patch trajectories in *Figure 3a*. The total equivalent strain map (*Figure 5b*) shows that the total equivalent strain in this area is above 1.4%, with the rest of the field remaining below 1.2% deformation.

The result in *Figure 5* is based on the alignment using a grid of 12×8 patches, which can track the local strain patterns with fine granularity. Reducing the number of patches can smooth small-scale fluctuations and reduce apparent peak strain values, but may under-sample genuine local distortions. This trade-off between patch resolution and deformation magnitude suggests that the optimal patch density may depend on sample type and exposure rate. *Unbend* allows users to tailor the number of patches to their specimen and exposure-fractionation scheme, optimizing alignment fidelity and desired amount of deformation correction.

## Quantitative measurement of corrected motion using 2DTM

To assess improvements in preserving high-resolution signal in the motion-corrected micrographs, we searched for large ribosomal subunits (LSUs) using 2DTM, as described in the Materials and methods section. Stronger high-resolution signal leads to both an increased number of detected targets (*Figure 6a*) and higher 2DTM SNR values (*Figure 6b*). We also compared the SNRs of the same detections present in both micrographs processed with full-frame alignment and those processed with local distortion correction. The SNR differences are shown in *Figure 6c*, sorted for better visualization.

In whole-cell *M. pneumoniae* samples, distortion correction resulted in a 4.2-fold increase in LSU detection (95.5 vs 22.9 detections per image), and the mean 2DTM SNR improved by ~3.1% (8.96±0.02 vs 8.69±0.01). The low values of standard error of the mean (SEM) (0.02 and 0.01) indicate a consistent 2DTM SNR improvement. In BS-C-1 whole-cell samples, the number of LSU detections increased from 55.9 to 69.1 (~23.6% improvement), and the 2DTM SNR improved from 9.88±0.02 to 10.42±0.02 (~5.5% improvement). For targets detected in both full-frame and local-distortion corrected micrographs, approximately 80% showed higher 2DTM SNR values after local distortion

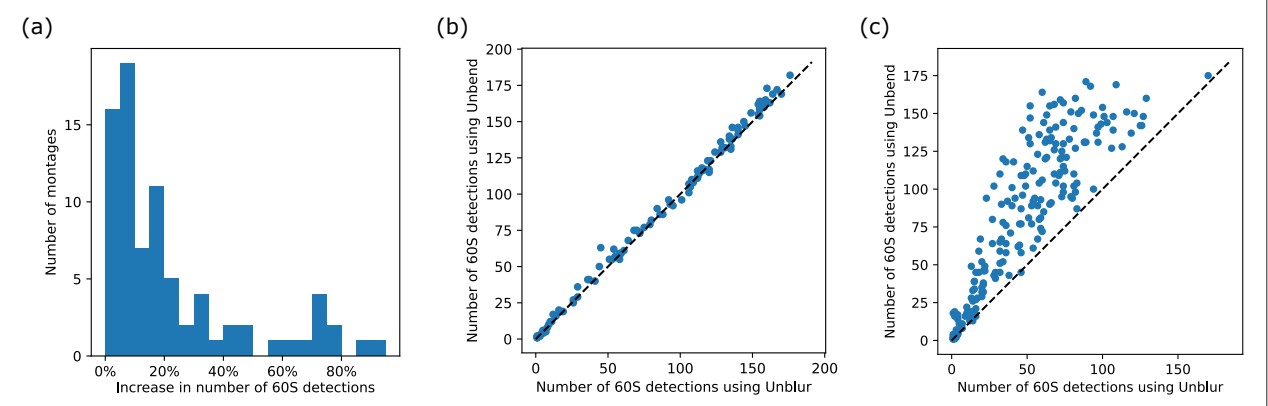

**Figure 7.** Impact of patch-based motion correction on the detection of 60S subunits in focused ion beam-milled lamellae analyzed in a large dataset. (**a**) Histogram showing the percentage of increase in 60S detection in montages processed using *Unbend*, compared to *Unblur*. (**b**) Scatterplot of the number of detected 60S per exposure in a representative montage that did not show a substantial increase of detection after processing with *Unbend*, compared to *Unblur*. (**c**) Similar to panel (**b**), but for a representative montage with a more substantial increase of detected targets after processing with *Unbend*, compared to *Unblur*.

correction (*Figure 6c*). Of the 20% of detected targets that displayed lower 2DTM SNR values after local distortion correction, some showed a reduction of up to a factor of 2. This reduction can be attributed to the motion correction process, which optimizes an overall loss function $L_2\left(K_x, K_y\right)$ and models motion using continuous functions to avoid patch alignments dominated by noise. This suggests that there are local areas that happen to be well aligned using full-frame alignment, and that end up with an overall worse alignment when additional degrees of freedom are added during local alignment. Nevertheless, local distortion correction resulted in a 317% increase in LSU detections in *M. pneumoniae* and a 24% increase for BS-C-1, with an overall 2DTM SNR improvement of 3–5%.

For lamellae prepared from *S. cerevisiae* and ER-HoxB8 cells, the mean number of LSU detections per micrograph increased from 350 to 401 (~15% increase) in yeast samples and from 8.5 to 10.0 (~17% increase) in mammalian samples. The mean 2DTM SNR for yeast samples improved from 10.18±0.01 to 10.62±0.01 (~4% increase), and for mammalian samples, it increased from 10.4±0.1 to 11.2±0.1 (~8% increase). As noted earlier, motion in ER-HoxB8 lamellae varies from less than 1 to 6 Å, contributing to a relatively larger SEM value (0.1).

Lysates generated from BS-C-1 cells showed no substantial increase in the number of LSU detections, but a 3% improvement in the 2DTM SNR was observed (from 13.84±0.04 to 14.25±0.04). As shown in *Figure 4*, cell lysates exhibit minimal local motion (mean shift amount 0.7 Å, *Figure 4—source data 1*), and full-frame alignment is sufficient to correct for most BIM in these samples. However, small amounts of local motion may remain, and this may explain the improved 2DTM SNR in 70% of the detected targets (*Figure 6c*).

## DeCo-LACE montages

To better understand the beam-induced deformations, we analyzed a dataset of micrographs collected from lamellae of *Candida albicans* cells (*Serrano et al., 2026*). In this dataset, lamellae of individual cells were imaged as montages (DeCo-LACE, *Elferich et al., 2022*). We performed motion correction of the individual exposures within these montages using either full-frame alignment or *Unbend*, and found that in 47 out of 82 montages, the number of detected targets increased by more than 10% (*Figure 7a*). This indicates that patch-based motion correction recovered signal that would otherwise have been lost to image blurring due to local deformations. In some montages, the number of detected targets was substantially higher in every exposure (*Figure 7c*), suggesting that these regions of the imaged lamella were generally prone to local deformations and the observed increase in the number of detected targets was not due to a few heavily distorted exposures.

For the remaining 35 montages that showed less than a 10% increase in target detection, the number of detections remained approximately unchanged between full-frame and local-distortion correction (*Figure 7b*). While we cannot rule out that areas in these lamellae experienced local

deformations that were not corrected by our algorithm, the overall similarity in the number of detections per exposure and in the 2DTM SNR of 60S targets compared to the full dataset suggests that these areas likely did not suffer from significant BIM. In fact, the mean 2DTM SNR in areas with less than a 10% increase in detections was 11.1, whereas the mean 2DTM SNR in areas with more than a 50% increase was 10.7. This suggests that patch-based motion correction recovers most, but not all, signal lost to local deformation.

In summary, our data suggest that local deformation in lamellae is not uniform: for reasons that remain unclear, some lamellae remain rigid while others display local deformation. Together with the finding that, despite patch-based motion correction, the average 2DTM SNR is still higher in areas without deformation, we propose that future work aimed at uncovering the physical basis of this variability could lead to improvements in in situ cryo-EM datasets.

## Comparison with published local motion correction methods using 2DTM

Several other movie alignment pipelines implement local motion correction using different deformation models. Widely used approaches include (1) quadratic (low-order polynomial) motion models, as implemented in *MotionCor2/MotionCor3* (*Zheng et al., 2017*), and (2) spline-based models that allow more flexible deformation fields, as implemented in *Warp* (*Tegunov and Cramer, 2019*) and *CryoSPARC* (*Punjani et al., 2017*).

In this section, we performed movie alignment using five methods (*Unbend*, *MotionCor2*, *MotionCor3*, *Warp*, and *CryoSPARC*) and then ran 2DTM on the resulting motion-corrected micrographs. Because several packages only support downsampling by integer factors and all movies were recorded at super-resolution, the aligned outputs were either not binned or binned by a factor of 2. To standardize sampling across methods and samples for downstream analysis, we then Fourier-binned the corrected micrographs to a final pixel size of 1.5 Å/pixel. For *Unbend*, all alignment parameters were kept at their default values. For the other packages, we matched the nominal model flexibility to *Unbend*'s model by using the same number of patches along x and y in *MotionCor2* and *MotionCor3*, and the same number of spline knots in *Warp* and *CryoSPARC*. All remaining parameters were left at their respective defaults to minimize method-specific tuning and reduce potential bias.

*Figure 8a* summarizes the number of 2DTM detections per micrograph for each motion-correction method. For particles detected by both *Unbend* and an alternative method, we computed the SNR difference as $\Delta SNR = SNR_{Unbend} - SNR_{other}$. The sorted $\Delta SNR$ distributions are shown in *Figure 8b*.

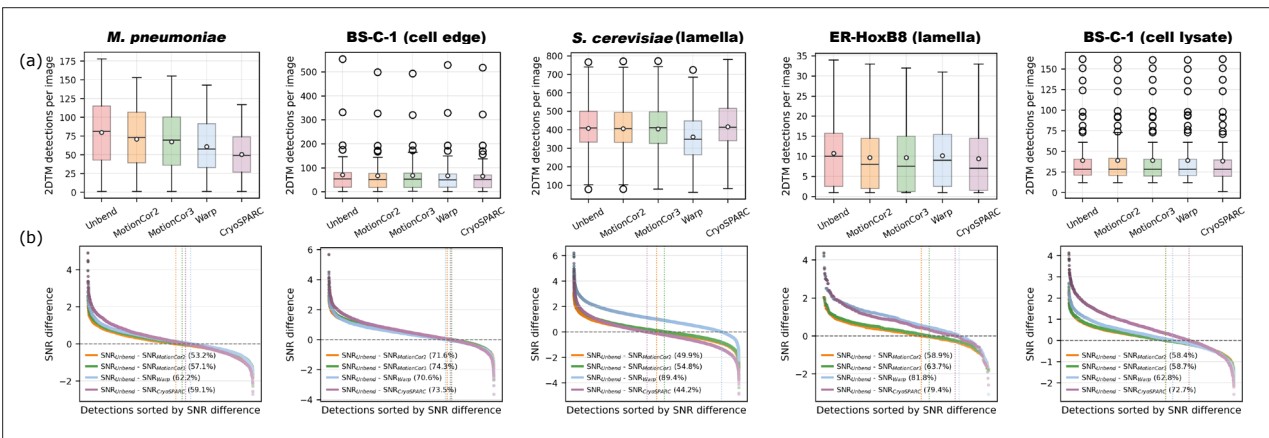

**Figure 8.** Comparison of local motion correction methods. (**a**) Box plots showing the distribution of the number of 2D template matching (2DTM) detections per micrograph for each motion correction method. (**b**) Distribution of 2DTM signal-to-noise ratio (SNR) differences between *Unbend* and the other tested methods for particles detected in common ($\Delta SNR = SNR_{Unbend} - SNR_{other}$). Vertical lines mark the zero-crossing in the sorted $\Delta SNR$ distribution for each method. Percentages in the legend indicate the fraction of commonly detected particles with higher SNR in *Unbend*-processed micrographs.

The online version of this article includes the following source data for figure 8:

**Source data 1.** Statistics table for detected particles from micrographs processed by different software.

For the *M. pneumoniae* sample, which exhibits the largest local motion (*Figure 4a*), *Unbend* produced the highest detection rate, with a mean of 79.7 detections per micrograph, followed by *MotionCor2* (70.6). In contrast, *CryoSPARC* produced 50.4 detections per micrograph, 36.8% fewer than *Unbend*. Consistent with this, among particles detected by both *Unbend* and *Warp*, over 60% had higher SNR in the *Unbend*-corrected micrographs.

For the cell edge sample, where the mean patch shift spans from <1 to >5 Å (*Figure 4b*), the mean detection counts between different methods are more similar than in the bacterial sample. However, the SNR comparison (*Figure 8b*) shows a clear advantage for *Unbend*: more than 70% of commonly detected particles have higher SNR in *Unbend*-processed micrographs than in micrographs processed by the other methods, indicating improved recovery of local signal.

For the yeast lamella sample, performance is more consistent between methods. Considering the SNR of commonly detected particles, *CryoSPARC* performs slightly better than *Unbend* (only ~44% of common particles show higher SNR in *Unbend*), *MotionCor2* performs similarly to *Unbend*, and *Warp* performs slightly worse (with our choice of run parameters). For the mammalian lamella sample, the methods yield similar detection counts (*Figure 8a*). However, the SNR comparisons favor *Unbend*: detections from *Unbend*-processed micrographs show higher 2DTM SNR than all other methods. In particular, *Unbend* yields higher SNR for ~80% of the particles common with the spline-based methods (*Warp* and *CryoSPARC*). For the cell-lysate sample, overlap in detections is high across methods, but 60–70% of commonly detected particles still show higher SNR in the *Unbend*-corrected micrographs.

Overall, *Unbend* provides the best and most consistent performance across datasets in our benchmark. *MotionCor2* and *MotionCor3* perform consistently and robustly across samples, whereas *Warp* and *CryoSPARC* show stronger dataset dependence. We aimed to minimize user bias by matching model flexibility and otherwise using default parameters; nevertheless, method-specific parameter optimization could affect absolute performance. A comprehensive, method-specific parameter benchmarking study is beyond the scope of this work.

## Unbend runtimes

We benchmarked the runtimes of *Unbend* using the movie datasets analyzed in this study. Each job was executed with 4 CPU threads on Intel Xeon Gold 5520+ processors. *Table 1* summarizes the runtimes together with key parameters of the input movies and the corresponding output micrographs. The runtimes range from ~1.5 to ~12.5 min per movie, depending primarily on the input

**Table 1.** Unbend runtimes on movies analyzed in *Figure 6*.

|  | *M. pneumoniae* | *Aethiops*: BS-C-1 (cell edge) | *S. cerevisiae* | *Musculus*: ER-HoxB8 | *Aethiops*: BS-C-1 (cell lysate) |
|---|---|---|---|---|---|
| Input movie size (pixels) | 5760×4092 | 5760×4092 | 11520×8184 | 11520×8184 | 11520×8184 |
| No. of frames per movie | 24 | 30 | 50 | 75 | 30 |
| Exposure per frame (e⁻/Å²) | 1.296 | 1.021 | 0.600 | 0.600 | 1.004 |
| No. of movies | 64 | 65 | 30 | 59 | 76 |
| Input pixel size (Å/pixel) | 1.053 | 1.33 | 0.53 | 0.415 | 0.415 |
| Defocus (μm) (mean ± standard deviation) | 1.3±0.5 | 1.5±0.3 | 0.4±0.1 | 1.0±0.5 | 1.2±0.4 |
| Patch number | 8×6 | 10×8 | 8×6 | 7×5 | 7×5 |
| Output image size (pixels) | 4044×2873 | 5107×3628 | 4070×2892 | 3187×2264 | 3187×2264 |
| Output pixel size (Å/pixel) | 1.5 | 1.5 | 1.5 | 1.5 | 1.5 |
| Runtime per movie (mean ± standard deviation) | 1m34s±5 s | 3m21s±35 s | 7m58s±26 s | 12m30s±43 s | 4m49s±9 s |

image size and the number of frames. *Unbend* is currently CPU-based and therefore slower than the GPU-accelerated implementations. The most time-consuming step is the pixel-wise interpolation, which is performed on unbinned frames. This explains why larger input dimensions and higher frame numbers result in longer runtimes. A GPU-accelerated version of Unbend, as well as additional code-level optimizations to reduce runtime, will be available in a future release.

## Discussion

In this work, we introduce *Unbend*, a cryo-EM movie alignment program that can perform local motion correction using a 3D cubic spline model. The program begins with full-frame alignment, followed by tracking of local motion via patch-based alignment. Our 3D cubic spline model is then fit to the patch motion and optimized by a summed patch-wise cross-correlation loss function, yielding refined model parameters. With this optimized model, pixel-wise shifts are calculated for each movie frame to generate the motion-corrected micrograph. *Unbend* is integrated into our *cis*TEM software, featuring a new GUI with real-time distortion visualization. Combined with our new program *shift_field_generation*, pixel-wise shift fields can be generated for calculating total and Von Mises equivalent strain. These metrics quantify the extent of deformation and can serve as a reference for adapting the number and overlap of patches to avoid introducing excessive errors in the deformation model.

### Model performance

We validated *Unbend* across diverse in situ samples, including whole cells, lamellae, and cell lysates, and used 2DTM to quantify the improvement in high-resolution signal in the motion-corrected micrographs. Across all sample types tested, *Unbend* boosted 2DTM SNR by 3–8% and the number of detected targets by up to 300%, with the largest gains observed in specimens exhibiting large local motion (whole cells and lamellae). While cell lysate showed minimal local motions, a 3% improvement in 2DTM SNR was still observed.

Equivalent strain analysis confirmed that, apart from some samples of the *M. pneumoniae* dataset, correction-induced deformations remain below 1.0%, suggesting that the model improves alignment without introducing excessive errors. The strain fields showing the maximum patch shift amounts in micrographs of different samples in *Figure 4* are presented in *Figure 4—figure supplements 1–5* and summarized in *Figure 4—source data 2*. For the *M. pneumoniae* sample in *Figure 4—figure supplement 1*, strain analysis revealed maximum Von Mises and total equivalent values exceeding 2.2% and 3.5%, respectively, consistent with plastic deformation. Comparison of the corresponding power spectra (*Figure 4—figure supplement 1a and b*) showed that the micrographs corrected with *Unbend* displayed stronger Thon rings at high spatial frequencies, confirming that the observed plastic deformation was, at least partially, accommodated.

Together, these results demonstrate that local motion correction is essential for obtaining fully motion-corrected micrographs, especially when imaging whole cells and lamellae.

### Implications and limitations

Our results indicate that specimen type and preparation method strongly influence local sample motion. Whole cells often exhibit large thickness gradients that can potentially create areas of differential motion under the electron beam, while in lamellae, which are generally thinner and more homogeneous, some stress may still be present due to the milling process and residual sample inhomogeneity. The local distortion correction model is particularly beneficial in this context, as it can compensate for these localized motion artifacts, leading to more accurate particle detection and improved 2DTM SNR. Cell lysates typically show less localized motion, and full-frame alignment is sufficient for detecting most targets using 2DTM, although some residual local distortion remains detectable and correctable.

Lamellae montages collected using DeCo-LACE (*Elferich et al., 2022*) further revealed substantial spatial heterogeneity in the amount of local motion, indicated by the range of 2DTM detection gains from under 10% to over 50% within a single sample. We found that montages with less motion better preserve high-resolution signal, suggesting that further improvements in the motion correction model, perhaps based on a more detailed physical mechanism, may yield additional benefits. For single particle samples, which are generally thin and homogeneous, local motion is less pronounced,

and full-frame alignment should be sufficient in most cases. Therefore, they were not tested in this work.

# Materials and methods

## Key resources table

| Reagent type (species) or resource | Designation | Source or reference | Identifiers | Additional information |
|---|---|---|---|---|
| Cell line (*M. pneumonia*) | M129 | *O'Reilly et al., 2020*; *Lucas et al., 2021* | ATCC 29342 | |
| Cell line (*C. aethiops*) | BS-C-1 | ATCC | CCL-26; RRIDs:CVCL_0607 | |
| Cell line (*M. musculus*) | ER-HoxB8 | *Elferich et al., 2022* | | |
| Cell line (*S. cerevisiae*) | BY4741 | *Lucas et al., 2022* | ATCC S288C | |
| Cell line (*C. albicans*) | PY6413 | *Serrano et al., 2026* | | |
| Software, algorithm | *cis*TEM | *Lucas et al., 2021*; *Grant et al., 2018*; *Elferich et al., 2024* | | https://cistem.org/ |

## Cell culture, cell lysate, and grid preparation

Adherent BS-C-1 cells were cultured as previously described by *Salgado et al., 2018*. Briefly, cells were maintained at 37°C in a humidified atmosphere with 5% $CO_2$ and grown in Dulbecco's Modified Eagle Medium (DMEM; Invitrogen) supplemented with 10% fetal bovine serum, 1× GlutaMAX (Thermo Fisher Scientific), and 1% penicillin-streptomycin (100 U/mL penicillin and 100 μg/mL streptomycin; Thermo Fisher Scientific). For grid seeding, cells were washed twice with pre-warmed PBS and detached using trypsin-EDTA (Gibco). Cells were counted, diluted to $1×10^5$ cells/mL and seeded onto fibronectin-coated (5 μg/mL in PBS) EM grids. BS-C-1 cells were obtained from ATCC and used without further authentication or testing for mycoplasma contamination.

Double-side glow-discharged, 200-mesh gold grids with a silicon oxide support film containing 2 μm holes and 2 μm spacing (Quantifoil) were used. Prior to use, grids were cleaned by three washes with ethyl acetate and exposed to UV light under a laminar flow hood for 45 min. Grids were placed into 35 mm glass-bottom tissue culture dishes (MatTek) for cell growth. Cells were allowed to adhere and spread for 24–48 hr before vitrification. Each grid was inspected under an inverted light microscope to confirm appropriate cell density and carbon film integrity. We aimed for approximately one cell per mesh; grids with excessive cell clumping were excluded.

Immediately before plunge freezing, grids were washed twice with warm PBS and transferred into Minimum Essential Medium Eagle Alpha (MEMα; Sigma) supplemented with 25 mM of HEPES buffer. Grids were then rapidly moved from the incubator to a Leica GP2 cryo-plunger; excess liquid was blotted from the back side for 8 s, and grids were plunge-frozen into liquid ethane cooled to –184°C.

Cell extracts were prepared from BSC-1 cells using digitonin-based semi-permeabilization. Briefly, cells were seeded in a 75 cm² flask and cultured to confluence. After washing twice with pre-warmed PBS, the cells were detached with trypsin-EDTA (Gibco) and collected by centrifugation at 300×*g* for 4 min. The cell pellet was resuspended in 100 μL of semi-permeabilization buffer (25 mM HEPES, 110 mM potassium acetate, 15 mM magnesium acetate, 1 mM DTT, 0.015% digitonin, protease inhibitor cocktail [1 tablet/10 mL; Roche], 40 U/mL RNase In [Promega], 1 mM EGTA) and kept at 4°C for 5 min. Following centrifugation at 1000×*g* for 5 min, the supernatant was collected for use in grid preparation without further modification. RNA concentration was quantified prior to grid preparation as a quality control measure, with no dilution of the extract performed. Glow-discharged Quantifoil 200-mesh gold grids with a silicon oxide support film containing 2 μm holes and 2 μm spacing were used. Plunge-freezing conditions used were identical to those described for whole cells.

## Cryo-EM data acquisition

Cryo-EM data were collected using a Titan Krios transmission electron microscope (Thermo Fisher Scientific) operated at 300 keV, equipped with a K3 direct electron detector and an energy filter (Gatan) set to a 20 eV slit width.

For cell edge data collection, medium magnification montages were used to select vitrified cell areas that were transparent enough for cryo-EM. Images were acquired at a nominal magnification of ×64,000 corresponding to a calibrated pixel size of 1.33 Å. A defocus range of –1.0 to –1.5 μm was targeted using SerialEM's autofocus function (*Mastronarde, 2005*) on a sacrificial area. Movies were recorded at an exposure rate of 1.02 e⁻/Å per frame to a total accumulated exposure of 30.6 e⁻/Å.

For the cell lysate, we used a nominal magnification of ×105,000 corresponding to a calibrated pixel size of 0.83 Å. We employed beam tilt to acquire multiple movies (5 per hole across nine holes) at each stage position. Zero-loss peak (ZLP) refinement was performed every 90 min at a unique location to avoid dark areas. Movies containing 30 frames, with an exposure of 1.0 e⁻/Å per frame, were recorded, resulting in a total exposure of 30 e⁻/Å.

## 2D template generation and 2DTM

For 2DTM, 3D templates were generated from trimmed atomic models containing only the LSU (50S or 60S), using the simulate program (*Himes and Grigorieff, 2021*). Specifically, models derived from PDB entries 5LZV (for mammalian samples: BS-C-1 and ER-HoxB8), 6Q8Y (for *S. cerevisiae*), and 3J9W (for *M. pneumoniae*) were used. The defocus of the micrographs was determined using CTFFIND5. 2DTM was performed using the *match_template* program (*Lucas et al., 2021*) as implemented in *cis*TEM. The search was conducted with an in-plane angular step of 1.5° and an out-of-plane step of 2.5°. For whole-cell and lamella samples, an additional defocus search was conducted within ±120 nm of the estimated defocus, using 20 nm increments.

Template-matched coordinates, Euler angles, and defocus values were extracted using the MT module within the *cis*TEM GUI and exported as .star files. These files were subsequently analyzed to quantify the number of detected targets and the 2DTM SNR using a custom Python package (https://github.com/LingliKong/2DTMSNR_MoCo_Bench; copy archived at *Kong, 2026*). The sample thickness was determined using CTFFIND5.

## Von Mises equivalent strain and total equivalent strain

The Von Mises equivalent strain derives from the Von Mises criterion, widely used in engineering and mechanics to predict yielding or failure of materials subjected to complex (multiaxial) stresses or strains. The Von Mises equivalent strain effectively measures how far a material is distorted from its original shape. In our case, with the shift field $\{u_x, u_y\}$, we can obtain the deformation gradient $\mathbf{F}$:

$$\mathbf{F} = \mathbf{I} + \nabla u = \begin{pmatrix} 1 + \dfrac{\partial u_x}{\partial x} & \dfrac{\partial u_x}{\partial y} \\ \dfrac{\partial u_y}{\partial x} & 1 + \dfrac{\partial u_y}{\partial y} \end{pmatrix}. \tag{17}$$

The Green-Lagrange strain tensor $\mathbf{E}$ filters out the rigid-body rotations, measures the true finite strains, and can be calculated by:

$$\mathbf{E} = \frac{1}{2}\left(\mathbf{F}^T\mathbf{F} - \mathbf{I}\right) = \begin{bmatrix} E_{xx} & E_{xy} \\ E_{xy} & E_{yy} \end{bmatrix}. \tag{18}$$

The total equivalent strain combines the magnitude of all deformation modes (axial+shear) and is expressed as:

$$\varepsilon_{total_{eq}} = \sqrt{E_{ij}E_{ij}} = \sqrt{E_{xx}^2 + E_{yy}^2 + 2E_{xy}^2}. \tag{19}$$

The Von Mises equivalent strain removes the volume change and only keeps the shape change:

$$\varepsilon_{vm_{eq}} = \sqrt{\frac{2}{3}E_{ij}^{dev}E_{ij}^{dev}} = \sqrt{\frac{2}{3}\left[\left(E_{xx} - \frac{E_{xx} + E_{yy}}{3}\right)^2 + \left(E_{yy} - \frac{E_{xx} + E_{yy}}{3}\right)^2 + 2E_{xy}^2\right]}, \tag{20}$$

where $E_{ij}^{dev}$ is the deviatoric strain:

$$E_{ij}^{dev} = E_{ij} - \frac{1}{3}\left(tr\,E\right)\delta_{ij} \tag{21}$$

where $\delta_{ij}$ is the Kronecker delta, and $trE$ is the volumetric part/trace, which represents the pure 'dilation' or 'compression' component that changes the area but not shape:

$$tr\,E = E_{xx} + E_{yy}. \tag{22}$$

## Acknowledgements

The authors thank members of the Grigorieff lab for helpful discussions. We are also grateful for the use of and support from the cryo-EM facility at UMass Chan Medical School. LK and NG acknowledge funding from the Chan Zuckerberg Initiative, grant #2021-234617(5022).

## Additional information

### Competing interests

Nikolaus Grigorieff: Reviewing editor, *eLife*. The other authors declare that no competing interests exist.

### Funding

| Funder | Grant reference number | Author |
| --- | --- | --- |
| Chan Zuckerberg Initiative | 2021-234617 (5022) | Lingli Kong<br>Nikolaus Grigorieff |
| Howard Hughes Medical Institute | | Ximena Zottig<br>Johannes Elferich<br>Nikolaus Grigorieff |

The funders had no role in study design, data collection and interpretation, or the decision to submit the work for publication.

### Author contributions

Lingli Kong, Conceptualization, Resources, Data curation, Software, Formal analysis, Validation, Investigation, Visualization, Methodology, Writing – original draft, Project administration, Writing – review and editing; Ximena Zottig, Resources, Data curation, Formal analysis, Validation, Investigation, Visualization, Writing – original draft, Writing – review and editing; Johannes Elferich, Conceptualization, Resources, Data curation, Software, Formal analysis, Validation, Investigation, Visualization, Methodology, Writing – original draft, Writing – review and editing; Nikolaus Grigorieff, Conceptualization, Resources, Software, Supervision, Funding acquisition, Validation, Visualization, Methodology, Writing – original draft, Project administration, Writing – review and editing

### Author ORCIDs

Lingli Kong ⓘ https://orcid.org/0000-0002-5808-2649
Ximena Zottig ⓘ https://orcid.org/0000-0002-2344-5163
Johannes Elferich ⓘ https://orcid.org/0000-0002-9911-706X
Nikolaus Grigorieff ⓘ https://orcid.org/0000-0002-1506-909X

Reviewer #1 (Public review): https://doi.org/10.7554/eLife.109119.3.sa1
Reviewer #2 (Public review): https://doi.org/10.7554/eLife.109119.3.sa2
Reviewer #3 (Public review): https://doi.org/10.7554/eLife.109119.3.sa3
Author response https://doi.org/10.7554/eLife.109119.3.sa4

## Additional files

### Supplementary files

MDAR checklist

## Data availability

Movies of samples not published previously have been deposited on EMPIAR: movies of ER-HOXB8 cells (EMPIAR-12944); movies of BS-C-1 cell edges (EMPIAR-12943); movies of BS-C-1 cell lysate (EMPIAR-12942). The source code for *Unbend* is part of *cis*TEM and available at https://github.com/timothygrant80/cisTEM/tree/unbend (**Grant, 2025**); binaries for most Linux distributions can be downloaded at https://cistem.org/downloads. The 2DTM detection and SNR comparison script is available at https://github.com/LingliKong/2DTMSNR_MoCo_Bench (copy archived at **Kong, 2026**).

The following datasets were generated:

| Author(s) | Year | Dataset title | Dataset URL | Database and Identifier |
|-----------|------|---------------|-------------|-------------------------|
| Kong L, Zottig X, Elferich J, Grigorieff N | 2025 | Unbend: Local correction of beam-induced sample motion in cryo-EM images using a 3D spline model | https://doi.org/10.6019/EMPIAR-12942 | EMPIAR, 10.6019/EMPIAR-12942 |
| Kong L, Zottig X, Elferich J, Grigorieff N | 2025 | Unbend: Local correction of beam-induced sample motion in cryo-EM images using a 3D spline model | https://doi.org/10.6019/EMPIAR-12943 | EMPIAR, 10.6019/EMPIAR-12943 |
| Kong L, Zottig X, Elferich J, Grigorieff N | 2025 | Unbend: Local correction of beam-induced sample motion in cryo-EM images using a 3D spline model | https://doi.org/10.6019/EMPIAR-12944 | EMPIAR, 10.6019/EMPIAR-12944 |

The following previously published datasets were used:

| Author(s) | Year | Dataset title | Dataset URL | Database and Identifier |
|-----------|------|---------------|-------------|-------------------------|
| Lucas BA, Himes BA, Xue L, Grant T, Mahamid J, Grigorieff N | 2021 | Locating Macromolecular Assemblies in Cells by 2D Template Matching with cisTEM | https://doi.org/10.6019/EMPIAR-10731 | EMPIAR, 10.6019/EMPIAR-10731 |
| Lucas BA, Zhang K, Loerch S, Grigorieff N | 2022 | In situ single particle classification reveals distinct 60S maturation intermediates in cells | https://doi.org/10.6019/EMPIAR-10998 | EMPIAR, 10.6019/EMPIAR-10998 |
| Elferich J, Plumb E, Grigorieff N, Arkowitz R | 2025 | Montaged cryo-electron micrographs of candida albicans cells | https://doi.org/10.6019/EMPIAR-12958 | EMPIAR, 10.6019/EMPIAR-12958 |

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
