## [Editor Report · eLife Assessment]

This paper describes Unbend - a new method for measuring and correcting motions in cryo-EM images, with a particular emphasis on more challenging in situ samples such as lamellae and whole cells. The method, which fits a B-spline model using cross-correlation-based local patch alignment of micrograph frames, represents an **important** tool for the cryo-EM community. The authors elegantly use 2D template matching to provide **convincing** evidence that Unbend outperforms the previously reported method of Unblur by the same authors. Comparison to alternative programs for motion correction shows smaller gains, but with interesting differences between data sets.

---

## [Referee Report · Reviewer #1 (Public review)]

Kong et al.'s work describes a new approach that does exactly what the title states, "Correction of local beam-induced sample motion in cryo-EM images using a 3D spline model." It is, therefore, a more elaborate approach than current methods in the field for the "movie alignment" stage. Additionally, the work uses 2DTM (2D Template Matching)-related measurements to quantify the improvement of the new method compared to other methods in the field. I find both parts very compelling (the new method and the testing approach)

On a "focused" view, the strengths of the work rest on presenting a better approach for motion correction and on measuring their performance very well at the 2D level in a compelling manner

On a more "general" view, the authors introduce the important notion that even one of the most worked-out steps in the processing workflow can still be done better in a measurable way, and that this could lead to better results beyond the 2DTM metrics used for testing, reflecting in better results along the processing pipeline (although the manuscript does not explore further this notion)

On the "usability" side, the method is still CPU-based and is slower than standards in the field. This may pose significant limitations in practical work, although the authors are aware of this issue and are working on it.

---

## [Referee Report · Reviewer #2 (Public review)]

Summary:

The authors present a new method, Unbend, for measuring motion in cryo-EM images, with a particular emphasis on more challenging in situ samples such as lamella and whole cells (that can be more prone to overall motion and/or variability in motion across a field of view). Building on their previous approach of full-frame alignment (Unblur), they now perform full-frame alignment followed by patch alignment, and then use these outputs to generate a 3D model of the motion. This model allows them to estimate a continuous, per-pixel shift field for each movie frame that aims to better describe complex motions and so ultimately generate improved motion-corrected micrographs. Performance of Unbend is evaluated using the 2D template matching (2DTM) method developed previously by the lab, and results are compared to using full-frame correction alone and to the leading local motion correction methods. Several different in situ samples are used for evaluation covering a broad range that will be of interest to the rapidly growing in situ cryo-EM community.

Strengths:

The method appears an elegant way of describing complex motions in cryo-EM samples and the authors present sound data that Unbend generally improves SNR of aligned micrographs as well as increases detection of particles matching the 60S ribosome template when compared to using full-frame correction alone and since review to the leading local motion correction methods. The authors also give interesting insights into how different areas of a lamella behave with respect to motion by using Unbend on a montage dataset collected previously by the group. There is growing interest in imaging larger areas of in situ samples at high resolution and these insights contribute valuable knowledge. Additionally, the availability of data collected in this study through the EMPIAR repository will be much appreciated by the field.

Weaknesses:

A major weakness was comparing this method to full-frame approaches only but this has since been addressed by the authors during review and Unbend is compared to MotionCor2, 3, CryoSPARC and Warp. The improvements here are smaller, generally it seems to perform on par with the above methods, but there are significant gains for certain samples (e.g. the M. pneumoniae sample). A comment from this reviewer about using an adaptive approach to decide if/when to proceed to the full Unbend pipeline, over full-frame alone, has been addressed by the authors.

---

## [Referee Report · Reviewer #3 (Public review)]

Summary

Kong and coauthors describe and implement a method to correct local deformations due to beam induced motion in cryo-EM movie frames. This is done by fitting a 3D spline model to a stack of micrograph frames using cross-correlation-based local patch alignment to describe the deformations across the micrograph in each frame, and then computing the value of the deformed micrograph at each pixel by interpolating the undeformed micrograph at the displacement positions given by the spline model. A graphical interface in cisTEM allows the user to visualise the deformations in the sample, and the method is proved to be successful by showing improvements in 2D template matching (2DTM) results on the corrected micrographs using five in situ samples.

Impact

This method has great potential to further streamline the cryo-EM single particle analysis pipeline by shortening the required processing time as a result of obtaining higher quality particles early in the pipeline, and is applicable to both old and new datasets, therefore being relevant to all cryo-EM users.

Strengths

(1) The key idea of the paper is that local beam induced motion affects frames continuously in space (in the image plane) as well as in time (along the frame stack), so one can obtain improvements in the image quality by correcting such deformations in a continuous way (deformations vary continuously from pixel to pixel and from frame to frame) rather than based on local discrete patches only. 3D splines are used to model the deformations: they are initialised using local patch alignments and further refined using cross-correlation between individual patch frames and the average of the other frames in the same patch stack.

(2) Another strength of the paper is using 2DTM to show that correcting such deformations continuously using the proposed method does indeed lead to improvements, as evidenced by the number of particles found and the quality of the detections (measured using 2DTM SNR). This is shown using five in situ datasets, where local motion is quantified using statistics based on the estimated motions of ribosomes. The same analysis is performed using other deformation correction tools, with Unbend showing superior performance in terms of particle detected or 2DTM SNR of the detections.

---

## [Author Response]

The following is the authors’ response to the original reviews.

We thank the reviewers for their constructive comments. A central concern raised is the comparison of performance with existing motion-correction methods. In response, we performed motion correction using several widely used approaches and compared results using the number of particles detected by 2DTM and their associated SNR. To minimize potential bias, we selected parameters to give each method a comparable level of model flexibility so that the results are as directly comparable as possible. Overall, Unbend performs the best. We note that extensive, method-specific parameter optimization could further affect absolute performance, and a comprehensive benchmarking study is therefore beyond the scope of this work

**Public Reviews:**

**Reviewer #1 (Public review):**
Kong et al.'s work describes a new approach that does exactly what the title states: "Correction of local beam-induced sample motion in cryo-EM images using a 3D spline model." I find the method appropriate, logical, and well-explained. Additionally, the work suggests using 2DTM-related measurements to quantify the improvement of the new method compared to the old one in cisTEM, Unblur. I find this part engaging; it is straightforward, accurate, and, of course, the group has a strong command of 2DTM, presenting a thorough study.However, everything in the paper (except some correct general references) refers to comparisons with the full-frame approach, Unblur. Still, we have known for more than a decade that local correction approaches perform better than global ones, so I do not find anything truly novel in their proposal of using local methods (the method itself- Unbend- is new, but many others have been described previously). In fact, the use of 2DTM is perhaps a more interesting novelty of the work, and here, a more systematic study comparing different methods with these proposed well-defined metrics would be very valuable. As currently presented, there is no doubt that it is better than an older, well-established approach, and the way to measure "better" is very interesting, but there is no indication of how the situation stands regarding newer methods.Regarding practical aspects, it seems that the current implementation of the method is significantly slower than other patch-based approaches. If its results are shown to exceed those of existing local methods, then exploring the use of Unbend, possibly optimizing its code first, could be a valuable task. However, without more recent comparisons, the impact of Unbend remains unclear.

We thank the reviewer for this important point. We agree that comparing against modern local motion-correction approaches is a valuable task. To address this, we added a new benchmarking section (pp. 17–18, lines 444–492, Fig. 8, Fig. 8—figure supplement 1) that compares Unbend against widely used patch-based local correction methods, including MotionCor2, MotionCor3, Warp, and CryoSPARC. Using the same 2DTM-based metrics described in the manuscript (detections per micrograph and SNR distributions for commonly detected particles), we find that Unbend provides the most stable performance across the tested datasets and, in most cases, yields higher detection counts and higher SNR than the alternative methods.

Regarding runtime, the current implementation is CPU-based and is therefore slower than some optimized GPU-accelerated packages. We now clarify this limitation in the manuscript (line 498–499). Our primary goal in this study is to improve motion-correction accuracy and quantify its impact using 2DTM-based measures. Importantly, higher-quality motion-corrected micrographs can reduce downstream processing cost (e.g., by increasing particle detection efficiency and reducing ambiguous candidates), so modest additional compute times at the motion-correction stage can be offset later in the workflow. We also note that GPU acceleration and additional code-level optimizations are planned for future releases (line 501-503); however, they are not required to evaluate the methodological contribution and the benchmarking results presented here.

**Reviewer #2 (Public review):**
Summary:The authors present a new method, Unbend, for measuring motion in cryo-EM images, with a particular emphasis on more challenging in situ samples such as lamella and whole cells (that can be more prone to overall motion and/or variability in motion across a field of view). Building on their previous approach of full-frame alignment (Unblur), they now perform full-frame alignment followed by patch alignment, and then use these outputs to generate a 3D cubic spline model of the motion. This model allows them to estimate a continuous, per-pixel shift field for each movie frame that aims to better describe complex motions and so ultimately generate improved motion-corrected micrographs. Performance of Unbend is evaluated using the 2D template matching (2DTM) method developed previously by the lab, and results are compared to using full-frame correction alone. Several different in situ samples are used for evaluation, covering a broad range that will be of interest to the rapidly growing in situ cryo-EM community.Strengths:The method appears to be an elegant way of describing complex motions in cryo-EM samples, and the authors present convincing data that Unbend generally improves SNR of aligned micrographs as well as increases detection of particles matching the 60S ribosome template when compared to using full-frame correction alone. The authors also give interesting insights into how different areas of a lamella behave with respect to motion by using Unbend on a montage dataset collected previously by the group. There is growing interest in imaging larger areas of in situ samples at high resolution, and these insights contribute valuable knowledge. Additionally, the availability of data collected in this study through the EMPIAR repository will be much appreciated by the field.

Thank you for this positive assessment.

Weaknesses:While the improvements with Unbend vs. Unblur appear clear, it is less obvious whether Unbend provides substantial gains over patch motion correction alone (the current norm in the field). It might be helpful for readers if this comparison were investigated for the in situ datasets. Additionally, the authors are open that in cases where full motion correction already does a good job, the extra degrees of freedom in Unbend can perhaps overfit the motions, making the corrections ultimately worse. I wonder if an adaptive approach could be explored, for example, using the readout from full-frame or patch correction to decide whether a movie should proceed to the full Unbend pipeline, or whether correction should stop at the patch estimation stage.

We thank the reviewer for suggesting an adaptive criterion to decide whether to proceed patch alignment or not. We agree that such an approach could be valuable for efficiency and for avoiding unnecessary model flexibility. However, our results indicate that a simple criterion based on the magnitude of estimated local patch motion is unlikely to be sufficient. For example, in the BS-C-1 cell lysate dataset, (see line 412-417 on page 16), we observe minimal local motion (Figure 4b) with mean patch shifts of only 0.7Å and full-frame alignment already yields comparable detection counts, yet local correction still produces a measurable SNR gain (13.84 ± 0.04 to 14.25 ± 0.04, 3%) and improves SNR for ~70% of the commonly detected targets (Figure 6c). This suggests that residual local distortion can remain even when overall local motion appears small. Establishing a robust, dataset-agnostic stopping rule would therefore require a dedicated, systematic benchmarking study across many samples and acquisition conditions.

**Reviewer #3 (Public review):**
SummaryKong and coauthors describe and implement a method to correct local deformations due to beam-induced motion in cryo-EM movie frames. This is done by fitting a 3D spline model to a stack of micrograph frames using cross-correlation-based local patch alignment to describe the deformations across the micrograph in each frame, and then computing the value of the deformed micrograph at each pixel by interpolating the undeformed micrograph at the displacement positions given by the spline model. A graphical interface in cisTEM allows the user to visualise the deformations in the sample, and the method has been proven to be successful by showing improvements in 2D template matching (2DTM) results on the corrected micrographs using five in situ samples.ImpactThis method has great potential to further streamline the cryo-EM single particle analysis pipeline by shortening the required processing time as a result of obtaining higher quality particles early in the pipeline, and is applicable to both old and new datasets, therefore being relevant to all cryo-EM users.Strengths(1) One key idea of the paper is that local beam induced motion affects frames continuously in space (in the image plane) as well as in time (along the frame stack), so one can obtain improvements in the image quality by correcting such deformations in a continuous way (deformations vary continuously from pixel to pixel and from frame to frame) rather than based on local discrete patches only. 3D splines are used to model the deformations: they are initialised using local patch alignments and further refined using cross-correlation between individual patch frames and the average of the other frames in the same patch stack.(2) Another strength of the paper is using 2DTM to show that correcting such deformations continuously using the proposed method does indeed lead to improvements. This is shown using five in situ datasets, where local motion is quantified using statistics based on the estimated motions of ribosomes.

Thank you for this positive assessment.

Weaknesses(1) While very interesting, it is not clear how the proposed method using 3D splines for estimating local deformations compares with other existing methods that also aim to correct local beam-induced motion by approximating the deformations throughout the frames using other types of approximation, such as polynomials, as done, for example MotionCor2.

We thank the reviewer for this suggestion. We agree that positioning Unbend relative to existing local motion-correction methods is important. In the revised manuscript, we added a dedicated benchmarking section comparing Unbend with widely used local correction approaches, including MotionCor2, MotionCor3, Warp, and CryoSPARC, using the same 2DTM-based metrics (Fig. 8, Fig. 8—figure supplement 1). This section is included on pp. 17–18, lines 444–492. To make the comparison as fair as possible, we matched nominal model flexibility across methods and otherwise used default parameters to reduce method-specific tuning. This expanded comparison provides a direct baseline against current patch-/spline-based approaches and shows that Unbend performs consistently across the in situ datasets evaluated here, with improvements in detection counts and/or SNR in multiple cases.

(2) The use of 2DTM is appropriate, and the results of the analysis are enlightening, but one shortcoming is that some relevant technical details are missing. For example, the 2DTM SNR is not defined in the article, and it is not clear how the authors ensured that no false positives were included in the particles counted before and after deformation correction. The Jupyter notebooks where this analysis was performed have not been made publicly available.

We agree that these technical details improve clarity and reproducibility. We have therefore made three changes.

(1) Definition of 2DTM SNR. We added an explicit definition of the 2DTM SNR in Section “2DTM provides a one-step verification for motion correction”, pp. 11, lines 277–287. Briefly, at each image location we compute cross-correlation values over the searched orientation space and define the 2DTM SNR as the maximum per location z-score across orientations.

(2) False-positive control / detection threshold. We clarified how detection thresholds were set to control false positives (pp. 11, lines 285–287). Specifically, we used the standard 2DTM statistical framework in which the threshold is chosen using the one-false-positive (1-FP) criterion (or equivalently, a specified expected false-positive rate). We applied the same thresholding procedure consistently across all motion-corrected micrographs. This ensures that particle counts before/after correction reflect changes in signal recovery.

(3) Reproducibility of the analysis. We have made the script used for the benchmarking and figure generation publicly available (pp. 24 line 622-623), and we provide a link in the Data Availability statement (pp. 25 line 650). The repository includes sample .star files and a python package that computes detections per micrograph, commonly detected particles, and SNR comparisons.

(3) It is also not clear how the proposed deformation correction method is affected by CTF defocus in the different samples (are the defocus values used in the different datasets similar or significantly different?) or if there is any effect at all.

We thank the reviewer for raising this point. In the revised manuscript, we now report the defocus ranges used for each dataset (Table 1) and clarify that all motion-correction comparisons were performed within each dataset using the same CTF estimation and 2DTM settings (pp. 23 line 615-618). Across the five datasets, four were collected at similar defocus ranges (1.0 µm to 1.5µm), whereas one dataset includes near-focus (0.4 µm) micrographs (Table 1). Because Unbend operates on frame alignment/warping rather than CTF modeling, we do not expect a defocus specific effect beyond indirect influences through image SNR and reliability of cross-correlation-based alignment.

**Recommendations for the authors:**

**Reviewer #1 (Recommendations for the authors):**
The obvious recommendation would be to use their 2DTM approach for a comparison of their new method with other currently used ones

We agree and added a new comparison section (pp. 17–18, lines 444–492). Addressed above in Response to Reviewer #1 Public Review.

**Reviewer #2 (Recommendations for the authors):**
(1) Line 29, typo. 3 ~ 8% > 3 - 8%.

Corrected.

(2) Lines 220 and 226. Should this be e-/Angstrom squared for the exposure?

Corrected to e^-^/Å^2^ (Now pp. 9 lines 230, 236).

(3) Figure 2 c-d. These are good for instinctively seeing the movement, but I found the legend confusing, as a 10 x 10 pixel array is mentioned, yet the schematics show a higher sampling (30 x 30 pixels? in c-e).

Thank you for pointing this out. The “10×10” annotation refers to the physical scale, whereas the grid represents pixel sampling. We removed the “10×10” label and now show only the pixel grid to avoid confusion. The caption has been updated to state that the grid corresponds to a 30×30 pixel sampling. (Fig. 2c, d; pp. 31, line 766)

(4) Figure 4. It would be good if the n of movies analyzed was given in the figure legend.

Thank you for noticing this. We report the number of movies per dataset in the corresponding summary table (Table 1).

(5) Figure 5. X/Y axes labels missing (assume pixels). Also, suggest changing the strain scale to % to match the main text description of this figure.

We added X/Y axis labels, changed the strain scale to % (Figure 5), and specified that the strains are per pixel on pp. 14 line 367. Correspondingly, the X/Y labels and strain scale in strain plots in Figure 4—figure supplementary 1 to 5 are also changed.

(6) Unify labelling of Figure 4 and 6 (i.e., Bacteria vs. M. pneumoniae, etc.).

Corrected. Sample labels are now consistent across figures. (Figures 4 and 6)

**Reviewer #3 (Recommendations for the authors):**
Some recommendations related to the points mentioned in the 'Weaknesses' section in the public review:(1) If feasible, it would be useful to see a comparison with other existing methods that estimate local deformations (e.g., MotionCor2), at least on some of the datasets. For example, does the proposed method lead to better 2DTM SNR in the detected particles compared to other methods, or higher detection numbers? Alternatively, if such a comparison would require too much additional work and the authors have good reasons to believe that the results are evident, it would be helpful to include a discussion about why the proposed method is expected to perform better, both in terms of the general approach and specific implementation details.

We agree that this comparison is important. (pp. 17–18, lines 444–492). Addressed above in Response to Reviewer #3 Public Review (1).

(2) It would be useful to define the 2DTM SNR in the main text of the paper, as well as to address the point about false positives in the picked particles.

We added an explicit definition of 2DTM SNR and clarified the detection thresholding/false-positive control used in our analysis (pp. 11, lines 277–287). Addressed above in Response to Reviewer #3 Public Review (2.1 and 2.2).

(3) Regarding the results shown in Figures 4 and 6: do the authors have any insight about how the CTF defocus affects the deformation estimation and correction across the different sample types?

We now report the defocus ranges used for each dataset (Table 1). We have addressed this problem in Response to Reviewer #3 Public Review (3).

(4) Will the Jupyter notebooks used for the 2DTM analysis be made publicly available?

Yes. We have deposited a python script used for the 2DTM benchmarking and figure generation in a public repository and added the link in Data Availability statement. (pp. 23 line 622, pp. 25 line 650). Addressed above in Response to Reviewer #3 Public Review (2.3).

(5) I would also appreciate a few words about the implementation details of the 3D spline model (e.g., what libraries have been used, if any, or if the authors have implemented their own code for this).

The 3D spline model and warping code were implemented by us (no external spline library was used) and the relevant implementation details are described in the “Sample distortion modeling and correction” section (pp. 7–10, lines 174–246). For optimization, we used the L-BFGS implementation provided by the dlib library, which is now explicitly cited (pp. 10, line 264).

Some comments regarding the presentation of the work:(1) I found the mathematical background on splines on pages 7-9 a little distracting from the main ideas of the paper, and I believe it could be moved to the methods section. A short description of this in the main text of the paper would suffice, and it would be useful to state clearly when this is background material and when it is the authors' contribution.

We appreciate the suggestion. Because Unbend includes an in-house spline implementation (no external spline library) and it is the central part of this work, we retained the spline description to support reproducibility. (pp. 7–10, lines 174–246).

(2) More generally, I found the whole method very interesting, but understanding exactly what all the steps involved were was a bit cumbersome, as they are spread across different sections of the main text. I think it would be useful to have a dedicated section giving the exact steps taken in the algorithm, possibly pointing to the relevant section in the text for more details about each step. This could be, for example, in the form of an 'Algorithm' box or a flowchart.

We added an Algorithm box as Figure 2 supplement summarizing the end-to-end workflow and pointing to the relevant sections for details (Figure 2—figure supplement 1 Algorithm, pp. 4, line 96–103, pp. 32 line 799). This is intended to make the sequence of steps easier to follow.

(3) In Figure 3, panels (b) and (c), the difference between the two micrographs, before and after correction, is not very noticeable, particularly the Thon rings in the spectra. I don't know if this is due to the image quality in the paper or if a better example could be shown. For example, the differences are clear in some of the supplementary figures.

Thank you for the suggestion. We revised the figure by adding annotations to show the recovered Thon rings. This figure shows a vertex motion and is intended not only to show improvement but also to illustrate complex, spatially varying deformation patterns that motivate the 3D spline model (pp. 12, lines 304–308). The supplementary figures display those with highest motions in each sample type, thus the Thon rings for the motion corrected micrograph in higher frequency space look more obvious. We also refer readers to the supplementary examples where the differences are more pronounced (pp. 12, lines 310–312).